# Preparation and Properties of Sustainable Brake Pads with Recycled End-of-Life Tire Rubber Particles

**DOI:** 10.3390/polym13193371

**Published:** 2021-09-30

**Authors:** Aitana Tamayo, Fausto Rubio, Roberto Pérez-Aparicio, Leticia Saiz-Rodríguez, Juan Rubio

**Affiliations:** 1Institute of Ceramics and Glass, Spanish National Research Council, Kelsen 5, 28049 Madrid, Spain; frubio@icv.csic.es (F.R.); jrubio@icv.csic.es (J.R.); 2Signus Ecovalor S.L., C/Caleruega 102, 28033 Madrid, Spain; rperez@signus.es (R.P.-A.); lsaiz@signus.es (L.S.-R.)

**Keywords:** composites, brake pads, end-of-life tire rubber, recycling, circular economy

## Abstract

Sustainable composite brake pads were processed by employing recycled end-of-life tire (ELT) rubber particles obtained by means of cryogenic grinding and ambient grinding. The effect of the grinding mechanism and concentration of ELT rubber particles was then reported. From the friction result test, better behavior in terms of coefficient of friction (COF) was obtained when 3% of ELT rubber particles were introduced into the composite. It was demonstrated that the size of the particles is not as determinant as the friction mechanism in the wear properties of the sustainable brake pads. Whereas, while increasing the ELT rubber particle size acts as detrimental to the COF either in the ambient or cryogenic grinding, at high friction distances, the better adhesion of the particles because of the rough surface of the particles subjected to ambient grinding enhances the long-life behavior of the composite brake pads.

## 1. Introduction

The road transportation sector is one of the most dynamic in economic development worldwide, but faces considerable challenges to preserve the environment. A cleaner transport implies not only reducing gas emission, but cleaner production technologies and waste recycling as well. Most parts of the rubber materials are employed as tires for passenger, off-road, and agricultural vehicles, trucks, airplane, or engineering vehicles, being around 70% of the overall rubber market allocated to tire production [1,2]. At the end of its life, the used tire becomes a waste that must be subjected to a waste management process according to the current environmental legislation. Here, the concept of Extended Producer Responsibility (EPR), which is defined in the European Waste Framework Directive 2008/98/EC3 (current 2018/851/CE [3]) as a tool to impose the producers of those products that become waste to consider them in the phases of prevention and management of the organization, is especially relevant. In this sense, the tire industry has been highly active in taking actions to organize the different players involved in the recovery chain by the creation of end-of-life tire (ELT) management systems at national and European level. These actions contribute significantly to the basic principles of the Circular Economy strategies in the use of wastes as resources, thus reducing the use of non-renewable natural resources [4].

ELTs are much more than waste. They are a source of material and energy resources as well in numerous market niches. Innovative solutions consisting of using these materials in diverse recycling applications like in polymer composites [5], concrete [6,7], asphalt [8,9] or as suitable media for biological growth and biofilm development in wastewater treatment systems [10] have been already reported in the literature. However, it is necessary to perform a deep investigation to establish new solutions to add value to these products.

One of the most compromising technological challenges that the rubber industry must face for recycling ELT is related to its complex structure and composition (vulcanized blend of different filled rubbers). Tires own complex designs possessing varied products [11] since their composition depends on their use and the type of vehicle. In all the cases, materials, manufacturing additives, and ingredients are combined in order to improve durability, performance, longevity, resistance to abrasion, etc. The typical composition of a tire is mainly based on four types of materials: natural and synthetic rubbers, carbon black or silica, structural materials (steel and textile fibers), and additives (in different low concentrations). All these components are placed in a special configuration within the tire (inner liner, tread, sidewalls, etc.) to conform the material, and then the final assembly is vulcanized for rubber curing (sulfur crosslinking) and transformation into a strong, elastic, rubbery, and hard state.

The complexity of the tire in some cases hinders the separation of all these materials once the tire has reached the end of its life, and consequently its overall recycling. Some of the processes involved are of less technological complexity, such as mechanical cutting, with several steps of size reduction (shredding and ambient/cryogenic grinding), but some others imply highly selective chemical reactions and complex physical treatments [12]. Thermo-mechanical (including those assisted by supercritical CO_2_ and/or devulcanizing agents), chemical, ultrasonic, microwave, and biological methods have been proposed as devulcanization methodologies [13].

The use of ELT rubber particles in brake pads turns out to be a smart strategy for a circular economy and tire recycling. Mutlu et al. reported the effect of the porosity of the ELT rubber on the wear properties of brake pads [14]. Through the addition of 5% to 15% wt. of ELT rubber particles, both the hardness and density of pads decreased, whereas the mean coefficient of friction (COF), specific wear, and porosity increased consistently [14]. Similarly, Singh et al. studied the substitution of brake pad fillers (barite) with 2.5% to 10% wt. of ELT rubber particles, obtaining the best stable COF and low fade when low rubber amounts were used. However, the best recovery, lowest wear, and low temperature rise occurred for high rubber contents [15]. 

In the present work, ELT rubber powder obtained by means of shredding and grinding were used in order to prepare sustainable brake pads for the transportation sector (trucks). Incorporating rubber particles in brake pads improves the friction properties during brake pressure [16]; thus, we used rubber particles obtained from ELT as a strategic component in sustainable brake pads with excellent performance. Both ambient and cryogenic processes were studied, with the focus on the optimization of the surface characteristics of the particles for a better performance of the sustainable brake pads. Ambient grinding is currently the most common method used in the market and produces irregular rubber particles with rough surfaces. In the ambient grinding process, the ELT particles pass through the nip gap of a shear mill or two-roll mill at room temperature. Grinding at temperatures below the rubber glass transition temperature (cryogenic grinding) provides smaller powders than ambient grinding and smoother surfaces. Therein, the particles are immersed in liquid nitrogen to convert them into brittle materials which are subsequently ground through a hammer mill.

To evaluate the properties of the brake pads, it must be taken into account that the incorporation of either synthetic or natural rubbers induces important changes in the wear and friction properties of brake pads. Thus, the substitution of the phenolic resin by increased amounts of styrene-butadiene-rubber (SBR) leads to a reduction of the hardness, flexural strength, and modulus, compressive modulus, and wear resistance, but the COF is usually enhanced. This behavior occurs because the friction properties are dominated by the type of polymeric binder used [17]. On the contrary, when the phenolic resin is replaced by nitrile-butadiene-rubber (NBR), a COF reduction occurs while the mechanical properties and wear resistance increase with the rubber amount [18]. Substitution of at least 10% of resin in the pad leads to a decrease of friction properties, although it favors wear properties and friction stability [18]. Besides, the size of rubber particles strongly influences the friction properties. When the used ELT rubber particles are of about 500 to 75 micrometers, it is reported that smaller particles lead to an increase in friction instability and higher wear rate, which is assigned to the contact plateau formed at the sliding interface [19]. On contrary, nanometer-sized SBR or NBR synthetic rubber nanoparticles induce a steadily variation of COF with temperature and the wearing rate is relatively low [20]. In most cases, the brake pads present COF values between 0.3 and 0.7 and wear rates between 3 × 10^−15^ and 45 × 10^−15^ mm^3^/Nm, which mainly depend on the type of abrasive inorganic particles [21], metallic components [22], and disposition and orientation of the fibers existing in the pad composition [23]. Taking all of this into consideration, the main objective of the present work was the preparation of sustainable brake pads by using recycled ELT rubber powder, assessing the use of different particle sizes obtained by ambient or cryogenic grinding in different concentrations in order to optimize the friction and wear properties of high COF brake pads.

## 2. Materials and Methods

### 2.1. Brake Pad Processing

The composite pad elements used in this study were prepared as monolithic pieces. The matrix was prepared by employing a phenolic novolac resin (F109, Sumitomo Bakelite Europe, Barcelona, Spain), Table 1, and several reinforcing components such as chopped glass fiber, graphite, steel, alumina and barite particles mixed in the relative proportions are given in Table 1. The composite pad element containing all the components are collected in Table 1, but no ELT rubber particles are labeled as Ref (or reference pad element) and it was used for comparison. The ELT rubber particles were subsequently incorporated to the matrix mixture constitutive of the reference pad in the concentrations specified in Table 2. Table 2 also specifies the characteristics and commercial names of the employed ELT particles. 

ELT rubber powder samples, named A80 and A60, were obtained by ambient grinding and supplied by Valoriza Medioambiente (Grupo Sacyr S.A., Madrid, Spain), and C20 and C40 were obtained by cryogenic grinding and supplied by Genan, Inc. (Viborgm, Denmark). All the rubber samples were composed of a crumb rubber mixture from all the different tire parts (inner liner, tread, sidewalls). The composite pad elements were labeled as “A” or “C” corresponding to ELT particles subjected to ambient or cryogenic grinding, followed with the particle size of the ELT rubber powder samples (as explained above) and ending with the concentration of the rubber particles in the composite. As an example, C40/03 and A60/20 samples correspond to the composite prepared with ELT rubber particles of <425 μm and <600 μm and 3 and 20% wt. concentration, respectively. 

The material processing methodology consisted of mixing the components of Table 1 plus the required concentration of the ELT particles to produce a warm, pressed green body which was subsequently post-cured. The mixing process of the powdered raw materials was carried out in a Turbula^®^ 3D shaker mixer (Willy A. Baschofen AG, Basel, Switzerland) for 30 min. Afterwards, the fibrous materials were added and mixed in the Turbula^®^ for another 5 min. The mixture was transferred to a stainless steel cylindrical preform and uniaxially pressed at 20 MPa for 1 min and at room temperature. The temperature was then increased to 180 °C while maintaining the pressure for 10 more minutes. The obtained green bodies were cooled back to room temperature and subjected to a post-curing treatment at 200 °C for 4 h in a conventional oven. The selection of this post curing temperature was conducted to avoid the undesirable degradation of the ELT rubber particles occurring at temperatures higher than 210 °C. According to the mold used, the size of the obtained pad elements were 34 mm in diameter and 5 mm in height. Specimens were prepared in duplicate. 

Particle size distributions (PSD) were determined by using a Mastersizer 3000 (Malvern Panalytical, Almelo, The Netherlands) instrument in dry medium to obtain the mean and nominal particle sizes, which are defined as those dimensions where the 50% wt. and 90% wt. of the PSD were below these given values, respectively. Density values were obtained by He pycnometry using an Accupyc 1330 (Micromeritics, Norcross, GA, USA) instrument. A Field Emission Scanning Electron Microscope (FE-SEM, Hitachi S4700, Tokyo, Japan) operating at 20 kV was used for the acquisition of the low- and high-magnification images of the ELT particles and brake pad elements. The samples were gold-sputtered to improve their conductivity prior to observation.

### 2.2. Brake Pads Characterization

Densities of the raw material (ρ) were determined by using a He gas pycnometer (AccuPyc II Micromeritics, Norcross, GA, USA) while pad element densities were obtained by the application of the water displacement Archimedes method. The porosity (in%) was calculated from the difference between the theoretical and real densities of the brake pads.

The tribological characterization was carried out in a tribometer (UMT3 CETR—Brucker, Campbell, CA, USA) using the Pin-On-Disc technique (Flat-on-Flat geometry) with a diameter of 8 mm. The pin friction material was made of stainless steel of 6.3 mm of contact and its rotation velocity was set to 10 Hz. Testing was carried out over 200 friction cycles consisting of a first pressure application of 2 N at 2000 rpm during 10 s, then the pressure was increased to 7 N to simulate a braking step for another 10 s while maintaining 2000 rpm. Afterwards, the pressure was decreased to 2 N in 3 s to achieve 0 rpm in order to simulate braking. Total distance covered was 250 m. During each test, Coefficient of Friction (COF) values were continuously recorded. The surrounding temperature was maintained between 25 ± 3 °C and humidity below 25%.

The wear rate (*W*) was obtained by using the following equation (Equation (1)):(1)$$W=\frac{V}{F\;d}$$
where *V* (in mm^3^) is the removed volume as measured from the worn profile, *F* is the applied force (in N), and *d* (m) is the sliding distance. The *V* values were determined from the corresponding friction profiles acquired by a contact profilometer Dektak XTL (Bruker, Billerica, MA, USA) with a vertical resolution of 0.1 μm.

A Specific Wear Rate test (*SWR*) was carried out by simulating a moderate braking condition in a lab scale. This test consisted of sliding the composite pad element on a cast iron disc at 200 rpm at 20 N. Pad elements and cast iron discs were of 34 mm and 220 mm in diameter, respectively. All brake samples were first weighed using an ultra-microbalance (±0.01 mg) before and after finishing the wear test cycles, which were carried out for 60 min each. Wear tests were repeated 8 times, implying a total of 8 h and more than 66,000 m of friction at 20 N. *SWR* were calculated in accordance with Equation (2):(2)$$SWR=\frac{M_{1}-M_{i}}{L_{i}\;P\;\rho }$$
(*i* = 1 to 8)
where *M_1_* and *M_i_* are the mass (g) of composite pad elements at initial (1) and *i-*cylce, respectively, *L_i_* is the friction distance (m) for the i-cycle, and *P* is the pressure (MPa) corresponding to the applied force (N) on the brake pad area (m^2^). ρ corresponds to the density of the composite brake pad element.

Prior to any tribological or friction test, all the samples were subjected to a polishing process using a SiC paper with grit sizes of 10.3 μm (P2000, Buelher, Düsseldorf, Germany) for 5 min in order to achieve more similar surfaces.

## 3. Results

As was commented before, commercial ELT rubber particles can be produced by ambient or cryogenic grinding, being the last method useful for obtaining low particle sizes. These processes lead to different particle surfaces, as shown in Figure 1. In Figure 1a,d, the low magnification image of the particles is shown, wherein the differences in size of the two types of particles, i.e., those which have been subjected to a cryogenic grinding (Figure 1a) and the particles subjected to the ambient grinding (Figure 1d), can be observed. For better visualization, particles of similar size were selected in the SEM images despite that they might not be the most populous ones, but their surfaces are representative of the whole. In Figure 1b, a typical particle of ELT after the cryogenic grinding is shown. Therein, it can be observed that the particles presented planar surfaces with well-defined edges. On contrary, the ELT particles obtained through ambient grinding, as shown in Figure 1e, were rougher and more irregular in shape than the former ones. 

Figure 1c,f shows high-magnification SEM images where the differences in the surfaces of the two types of particles are enhanced. The irregular ELT particles with rough surfaces obtained by ambient grinding thus possess a higher specific surface.

In Figure 1g, the PSD of the four types of particles used in this work is also shown. The differential distribution (left axis) allows for determining the mean particle size, whereas the cumulative distribution is commonly used to classify the ELT particles as a function of its nominal particle size. In should be noticed that in the case of the particles labeled C40, below 1% of particles were larger than 320 µm, and in the case of the C80, less than 5% of particles were higher than 200 µm. The differences in the mean particle size can be also appreciated in Figure 1a,d, as commented above.

In the formulation of the actual composite brake elements, the authors also emphasize that commercially available brake pads contain more than 30 constituents. Nevertheless, only seven ingredients were employed here in order to clearly identify the effect of the ELT rubber particle interactions with the remainder elements. In the following sections, the influence of concentration and particle size of ELT rubber in the friction and wear properties of high COF brake pad composite elements are described. 

### 3.1. Influence of the Concentration of ELT Rubber Powder 

Table 3 collects the corresponding real and theoretical densities determined in the pad composite elements prepared with the A60 particles, obtained through ambient grinding. It is obvious that the incorporation of ELT rubber particles leads to a decrease of both theoretical and real densities because of the lower density of rubber with respect to the other pad components and an increase in the pad porosity. This porosity appeared because of the rubber particle compression exerted during the pressing process that could lead to the formation of voids between the component particles that cannot be filled by the resin when it flows at high temperatures. In this Table 3, it is observed that the porosity increased progressively with the amount of ELT rubber particles, and this increase was more pronounced at rubber concentrations beyond 10%.

#### 3.1.1. Friction Characterization

Figure 2a shows an example of the friction test carried out for all the samples following the Pin-On-Disc test procedure (in this case, the first cycle of the reference material was represented, Ref). The three steps involved in the experiment cycle (initiation, acceleration, and braking) are clearly reproduced in the shape of the COF curve presented in Figure 2a, where smoother curves are reflected in the braking step because of the higher applied force. This cycle was repeated 200 times in each sample, and results obtained at 50, 100, 150, and 200 friction cycles are shown in Figure 2b.

As can be observed in Figure 2a, the COF followed a typical harmonic oscillation, which is probably due to the high rotation speed (2000 rpm) at which the friction tests were carried out. Similar behaviors have been reported for different brake pads in slip-stick tests when the friction velocities are higher than 2 mm/s for pads containing rubber particles [19] or higher than 11 mm/s for pads containing glass fibers [23]. When the test speed is low or because of the application of high friction forces, the stick-slip phenomenon dominates the performance [24]. Figure 2b also shows that the COF curves were almost overlapped in the different friction cycles for all ELT rubber particle concentrations. It is nevertheless appreciated that at large ELT concentrations, the COF was more dispersed when the number of cycles increased.

Figure 3 shows the surface profiles (z, in mm) of the friction test carried out for the different pad elements prepared with the A60 ELT rubber particles at different concentrations. These profiles correspond to an average value measured on four positions (at 0, 90, 180 and 270 degrees) perpendicular to the sliding direction. From the topography of the curves shown in Figure 3, it can be inferred the removal of some material during friction test occurred and, in some cases, ripped particles out of the friction surface. It is nevertheless clear that the material removal increased with the rubber concentration in the pad element, as a result of the low wear resistance of such particles. 

COF values determined form the curves of Figure 2b are collected in Table 4. In general, the COF values were mostly stable at different braking cycles; however, a small increase in the COF value with the sliding distance (or the number of braking cycles) and a small decrease with ELT rubber concentration were observed. The initial value of the COF determined in the Ref pad element (reference pad, with no ELT particles incorporated) was close to 0.56 and at the end of the experiment it reached a value of 0.63. In the meanwhile, the pad element A60/20, which contained 20% ELT rubber, showed an initial COF value close to 0.48 and ending at 0.55 after 200 Pin-On-Disc cycles. Similar results have been found for brake pads made of rubber nano- and micro-powders [19,20] and for different abrasive particles such as SiC, SiO_2_, and ZrSiO_4_ [21]. The COF values obtained in our work can be assigned to the high Mohs microhardness of the alumina particles used in this study (Mohs = 9) and are also similar to brake pad elements containing different concentration of metal fibers [25]. 

The wear resistance is a useful parameter to estimate the life in service of the brake pad. The determined *W* values corresponding to the profiles shown in Figure 3 are also given in Table 4. It is observed that the general trend is an increase in the wear volumes with the rubber concentration in the pad element, as corresponding to a low interfacial bonding between the rubber particles and the binding resin [19]. This phenolic resin used as binder presents a better surface interaction with the inorganic and metal components forming part of the composition of the brake pad element than with the rubber particles [26]. As Mutlu et al. [14] demonstrated, rubber concentrations between 5% and 10% can provide a beneficial effect in the specific wear, while they have a negative effect if the concentration is higher than 12.5%. On their side, Singh et al. [15] found a decrease in *W* with rubber concentration using rubber particles of 75 μm in size. In the prepared pad elements, a decrease of the *W* values for rubber concentrations between 5% and 10% was observed, but afterwards they increased, as Mutlu found. The discrepancies with the Singh’s work are attributed to the different particle size used.

#### 3.1.2. Specific Wear Rate (SWR)

Table 5 shows the recorded weight losses in the composite element pads containing 0% to 20% ELT rubber particles subjected to continuous braking cycles from 1 to 8 h. The general trend is an increase in the weight loss with both the amount of rubber in the pad elements time subjected to braking. During the whole braking period, a sustained weight loss in brake pad containing low ELT rubber contents occurred, whereas in those element pads with 10% ELT rubber particles and above, a sudden increase in the weight loss took place after 4 h of being subjected to braking cycles. 

The recorded weight losses collected in Table 5 were used to calculate the *SWR* according to Equation (2). Figure 4 shows the calculated *SWR* values in the studied element pads, which are in the range of 10^−14^ to 10^−13^ m^2^/N, in line with the *SWR* values claimed in commercial pads [27]. Similar *SWR* values are also reported for metallic, semi metallic, and non-asbestos organic brake pad elements [28] and in brake pads prepared with different phenolic resins [29]. These values, however, are very low compared with those found in some other pad compositions containing carbon black rubber materials [30], and even lower than those obtained for rubber materials reinforced with graphite [31] or layered silicates [32]. 

As it is observed in Figure 4, for rubber concentrations below 15%, the *SWR* values were practically constant over the whole braking distance of 66 × 10^3^ m, although a small decrease was observed during the first 2500 m. When the ELT rubber concentration was 15% or 20%, the *SWR* experiments showed a rapid increase after the first 30 × 10^3^ m and afterwards became stabilized again. The observed increase in the *SWR* with both the ELT rubber concentration and brake distance is in complete accordance with the *W* values given in Table 4 despite the fact that different experimental techniques were used to calculate the reported values.

### 3.2. Influence of the ELT Rubber Particle Size and Grinding Process

Real and theoretical densities of the brake pads obtained with different rubber particle sizes are collected in Table 6. It was observed that the ELT rubber particle size did not have any influence on the real density of the composite pad elements except when using particles of a high size (A80), which induced a small decrease in the density. According to these values, the main factor affecting the composite pad density and, thus the final porosity, was the rubber particle concentration, but not its size.

#### 3.2.1. Friction Characterization

The friction curves obtained for the composite pad elements prepared with ELT rubber with different particle sizes were similar to the curves shown in Figure 2, with a characteristic shape of harmonic oscillator especially evident in the composite pad prepared with the smallest ELT rubber particles (C20). Both the size of the ELT rubber particles and the grinding procedure determine the COF (Table 7), being the lowest COF values encountered for the composite pad elements subjected to the cryogenic grinding and bigger particle size. In the composite pad element containing the largest rubber particles and subjected to ambient grinding (A80), a decrease in the values of COF was also observed, and anomalous behavior was also observed by some other authors [19,20]. Moreover, as observed in Table 7, the general trend is an increase in the COF value with the sliding distance (number of cycles) in all the composite pad elements, except in the material labeled C40/03 fabricated with the particles sized <320 μm nominal particle size, where the COF became stabilized after a certain number of braking cycles. It should be also noticed that the biggest dispersion in the COF values was encountered in the brake pads containing the ELT rubber particles A60, which were subjected to ambient grinding and whose nominal particle size was below 470 μm.

Figure 5a shows the surface profiles after the friction test carried out for the composite pad elements prepared with ELT rubber particles of different sizes. In all curves, a down depression caused by material removal because of the pin contact during the test can be observed. From these profiles, the corresponding *W* values were calculated and are given in Table 7. As reported by Chang et al. [19], when rubber particle size decreased from 450 to 75 μm, a high shear force was induced after a stick episode when low particle sizes were used. However, Liu et al. found a relatively low wearing rate for rubber particles close to 0.1 μm, and the results were directly related to the dispersion of the particles in the pad mixture [20]. Therefore, in the prepared composites, it was deduced that the low wear value obtained for the composite pads containing the smaller rubber particles might be related with a better dispersion of the small particles in the pad element. 

In Figure 5b, the SEM image of the surface of the brake pad labeled C20/03 after being subjected to the pin-on-disk test is shown. For clarification, we highlighted, with a dotted line, the limit of the test (the bottom part of the image corresponds to the eroded area). Therein, some graphite particles can be observed, as planar, dark, and large particles, glass fibers, and ELT rubber particles are present in both parts of the image. In the eroded area, it is possible to observe some voids appearing because of the elimination of particles with the consequent debris formation as well as some remainders of the tribolayer formed during the test. The size of the voids was about 100–200 μm, which is approximately the size of the ELT rubber particles. In the images, the identification of the ELT rubber particles in the brake pad elements was realized by recognizing these particles with a weak or even hollow interfaces, which is the origin of the porosity in the pad elements. In Figure 5c, the high-resolution SEM image of an ELT rubber particle partially covered with a planar graphite particle is presented, and the poor interface between the matrix and the ELT rubber particle can be observed. 

#### 3.2.2. Specific Wear Rate (SWR)

Table 8 gives the recorded weight losses after different braking cycles for the composite pads containing 3% ELT rubber particles with particles of nominal sizes below 200, 320, 470, and 650 µm each. As it would be expected, the weight loss increased with the number of friction cycles due to a material removal effect. These results fall in the same range as the ones encountered for different rubber concentrations and those given in Table 5, but notable differences were found when using the different particle sizes. As commented before, not only the particle size influences the wear behavior of the composite pad elements, but also the particle grinding process. Either in the cryogenic grinding or the ambient grinding, the particles with the lower size in each case presented the highest weight loss after the braking cycles. 

The *SWR* values calculated from the weight losses presented above and by using Equation (2) are shown in Figure 6. The general trend is similar to the ones observed in Figure 4 for composite pad elements with rubber concentrations below 10%. Here, it was observed that except for the first 16,000 m, the *SWR* was practically constant, indicating a good stability of the friction properties of the composites. The smallest *SWR* values corresponded to the composite pad elements with rubber particles of below 650 µm (A80), while the higher were obtained for those with particle size of <200 µm and independent of the grinding process. This result is similar to Chang’s [19] work, which found an increase in *SWR* values as the particle size decreased and which attributed the observed behavior to the higher formation of wear debris in those pads prepared with the smallest particle size. In the prepared composite pad elements, the lower interaction of the rubber particles with other pad components may cause this higher *SWR* and therefore, in the same line, higher rubber concentrations or lower particle sizes lead to higher *SWR*.

## 4. Discussion

It is known that cryogenic grinding is more effective in producing fine particles than ambient grinding [33], as can be deduced from the particle size distributions shown in Figure 1. In these particle size distributions, the size span, which is defined as (d_90_ − d_10_)/(d_90_ + d_10_), was calculated, being d_10_ and d_90_, the size of the particles whose dimensions were the 10% wt. and 90% wt. of the PSD (notice that d_90_ corresponds to the nominal value). The encountered size span values were 0.57, 0.60, 0.50 and 0.42 for C20, C40, A60, and A80, respectively, indicating that these particles obtained through ambient grinding were less polydisperse than the ones obtained by the ambient grinding procedure. According to Voivrest et al. [34], polydispersity of granular media does not affect the shear strength, but it dominates the adhesion forces between particles. Additionally, the concentration of ELT rubber particles as well as their mean size were demonstrated to play a key role on the tribological properties of the composite brake pads. Chang et al. [19] reported a decrease of friction level with the particle size in brake pads prepared with 10% rubber particles, whereas Liu et al. [20] stated that by employing up to 5% of rubber nanoparticles, an improvement of the friction properties with the size of the particles was obtained. To study the effect of the concentration in the brake pads, the A60 particles were selected, possessing the medium size span (s = 0.50) to minimize the effect of the different packing density of the ELT rubber particles in the brake pads attributed to the different particle polydispersity. With these considerations, and based on the data presented in Figure 2b and collected in Table 4 and Table 7, we can extract that although the normal behavior was an increase of the COF with the number of braking cycles, this increase in the COF was most noticed in the brake pads containing 3% ELT rubber particles and when using the particles with the mentioned size (A60), as shown in Figure 7. Error bars were calculated taking into account the standard deviation within braking cycles in steps of 50 cycles.

Liew et al. [35] prepared a new brake pad material where the harmful asbestos component was replaced and compared the COF of the material with a commercial brake pad. In all the cases, the COF barely reached 0.5, which is almost the minimum value of the brake pads containing the A60 particle. Contrary to that which Liew et al. found, the COF increased with the sliding distance because of the different abrasion mechanism (plastic deformation attributed to the presence of the rubber particles). 

It is known that the properties of the particle-resin composites depend strongly on the particle size, particle–matrix interface adhesion, and particle loading. The particle-matrix interface is also determined by the size of the particles and their surface properties. A closer look into the results shown in Figure 7b allows distinguishing two differentiated behaviors dependent on the grinding procedure. Particles sized below 320 μm were subjected to a cryogenic grinding; thus, they were brittle and their surface was defined by planar sides, as shown in the SEM images in Figure 1. There was almost no dispersion between the different braking cycles in the COF. On contrary, the use of particles subjected to ambient grinding produced a major dispersion of the results, especially when using the particles of the lower size, a fact that is attributed to their inhomogeneous surface [36]. Although the A80 particles also possess a highly homogeneous surface, the decrease of the COF must be attributed to the plastic deformation of the ELT particles, as observed by some other authors [19,20]. Cryogenic grinding produces regular particles with a smooth surface and therefore a lower COF was observed [37]. Binding to the resin matrix is also enhanced in the case of the cryogenic grinding, a conclusion which can be extrapolated from the low data dispersion obtained along the different braking cycles [36]. 

At this point, it should be also taken into account that, for a brake system, the frictional power (*FP*) is an expression of the energy transformation resulted during braking (transformation to heat), and it is defined as a function of the sliding velocity, v, as follows:(3)FP=COF N v

*FP* is then related to the vehicle speed, the applied pressure (*N*) to the braking pad, and the quality of the friction pad or its COF value [38]. For a specified distance (number of braking cycles), *SWR* and *FP* (Equation (3)) can be calculated. The values of *SWR* and *FP* at the different braking distances in the tests carried out at medium speed (2.30 m/s) and medium friction load (20 N) which correspond to contact pressures (*p*) of 0.022 MPa are plotted in Figure 8. In all the cases, the braking area was of 9 cm^2^ in area and the *p.ν* values were 0.43 MPa m/s for all prepared composite pad elements that, despite being relatively low (they must be in the range 0.3 to 20 MPa m/s), some authors have considered adequate to investigate these materials [39]. The obtained SWR values were between 10^−4^ mm^3^/N for the pads containing the A80 particles and at high sliding distances to about 9 × 10^−4^ mm^3^/N for the brake pads containing the A60 particles at the maximum concentration. These values fall in the range of some brake pads containing natural recycled particles [40], where a SWR value of 7.22 10^−4^ mm^2^/N under a load application of 7.5 N has been reported, but they are higher than the composite pads filled with abrasive ceramics possessing COF of about 0.7 and SWR values below 4 10^−4^ mm^2^/N at 20 N load [41]. 

As observed in Figure 8, the *FP* range was very small since both N and *ν* of Equation (3) were the same values in all tests. A general trend of a decrease of *SWR* with *FP* when using the same particle sizes (Figure 8a) was noticed, similar to that observed before in some other works [28]. Nevertheless, the slope was quite different depending on the braking distance. At low braking distances, the *FP* values almost showed no variations with the *SWR,* whereas at braking distances beyond 41k, the *FP* tended to decrease rapidly with the *SWR*. As Wei et al. described [28], the wear and friction processes are determined by adhesion and abrasion mechanisms between the surfaces of contacting materials, i.e., the brake pad element and the cast iron disc. While adhesion varies with the sliding velocity and contact pressure, abrasion is independent of both parameters [28]. In this case, the abrasive mechanism dominates the friction process at low braking distance and then suddenly becomes mostly adhesive. This observation is attributed to the rough surface of the ELT particles obtained by ambient grinding. Independently of the concentration used, the ELT-matrix interface was quite poor and then after a certain number of cycles, the change in the fiction mechanism is attributed to the removal of the ELT particles. On the contrary, as shown in Figure 8b, when using the same concentration of particles but of different sizes, the abrasion mechanism seemed to dominate the friction characteristics. There, at low and high sliding distances, there was observed a slight increase of the *SWR* with the *FP*. The particles obtained through cryogenic grinding presented, in general, a lower COF and thus lower *FP*. Since the same concentration of particles was used, the increase of *SWR* with the *FP* should be attributed to different adhesion forces because of the different grinding procedures followed for obtaining the particles and therefore to a different particle-matrix interaction. This result is similar to the studies carried out by Zhang et al. [42], who demonstrated that the reaction between the different components of the brake pads because of the temperature increase at high friction cycles might contribute to the occurrence of the different abrasion mechanisms found as the sliding distance increases.

## 5. Conclusions

This work thus demonstrated the suitability to use ELT rubber particles in the manufacturing of sustainable composite brake pads with good performance characteristics. The optimum amount of ELT particles in the brake pads was about 3% wt. to achieve a high COF and long-life performance; nevertheless, this percentage will depend upon the particle size and the grinding procedure. Increasing the particle size of the ELT rubber caused a decrease of the COF and this value was lower in the case of the particles obtained through cryogenic grinding. Ambient grinding produced rougher particles than cryogenic grinding and therefore the rubber-matrix interface was enhanced. The abrasive friction mechanism dominated the behavior of the composite brake pads, a result that was more emphasized in the particles subjected to ambient grinding, whereas after a certain number of cycles, or braking distance, the friction mechanism was mostly adhesive independently of the grinding procedure. At the end, the grinding method determined the fictional properties of the pads and the ambient grinding enhanced the long-life behavior of the brake pads.

## Figures and Tables

**Figure 1 polymers-13-03371-f001:**
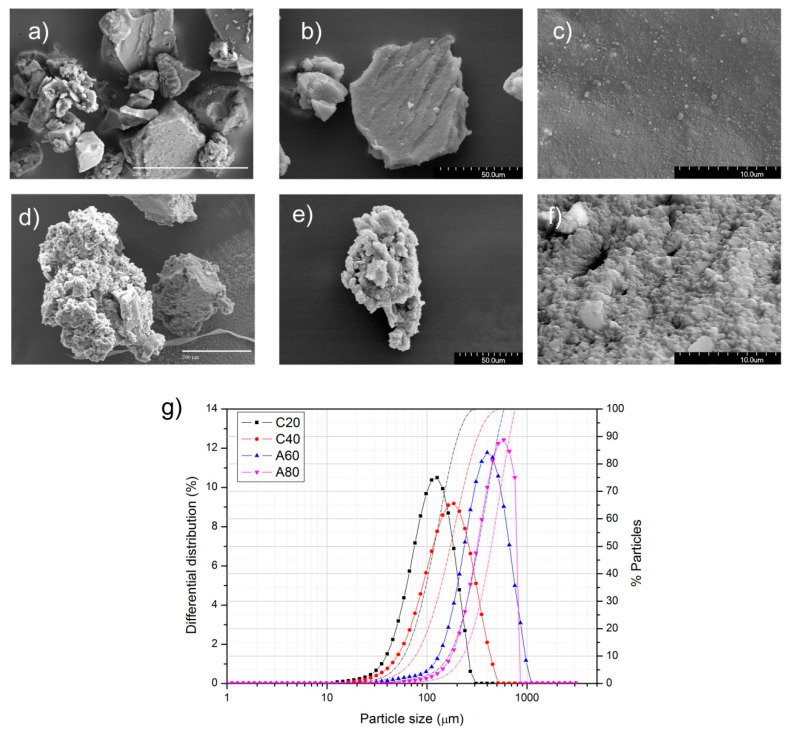
SEM images of the ELT particles obtained through (**a–c**) cryogenic grinding (sample C20) and observed at low (**a**,**b**) and high (**c**) magnification; SEM images of the ELT particles obtained through (**d**–**f**) ambient grinding (sample A60) and observed at (**d**,**e**) low and (**f**) high magnification; (**g**) differential particle size distribution (left axis) and cumulative particle size distribution (right axis) of the used ELT particles.

**Figure 2 polymers-13-03371-f002:**
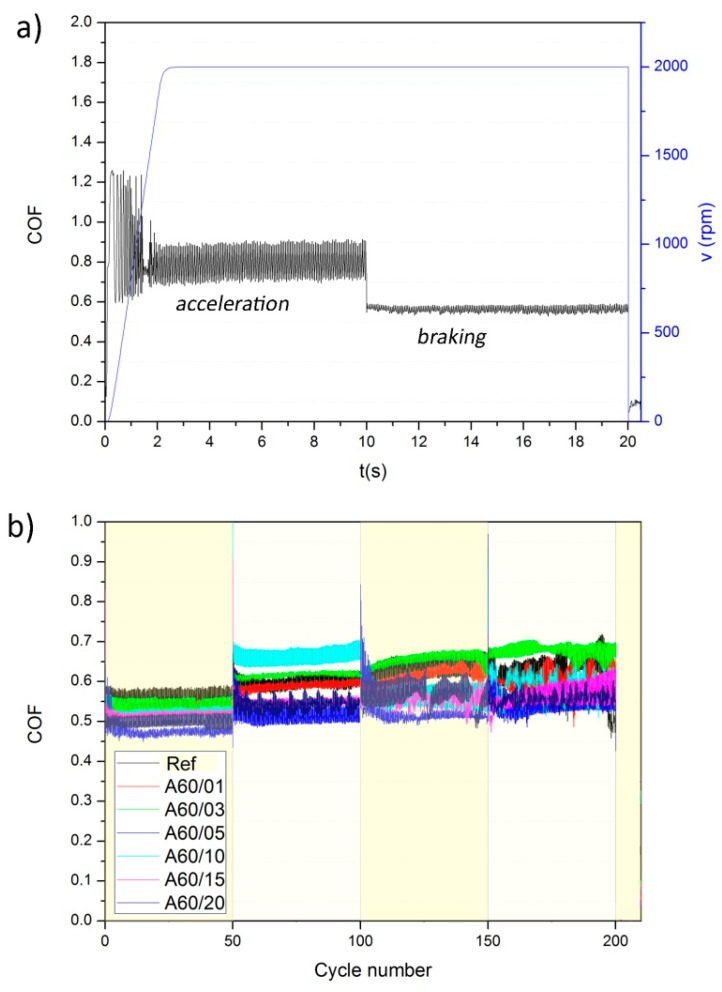
(**a**) Example of a generic friction curve obtained during the whole Pin-On-Disk experiment: COF and v(rpm), and (**b**) COF curves for different ELT rubber particle concentrations at 50, 100, 150, and 200 cycles of friction.

**Figure 3 polymers-13-03371-f003:**
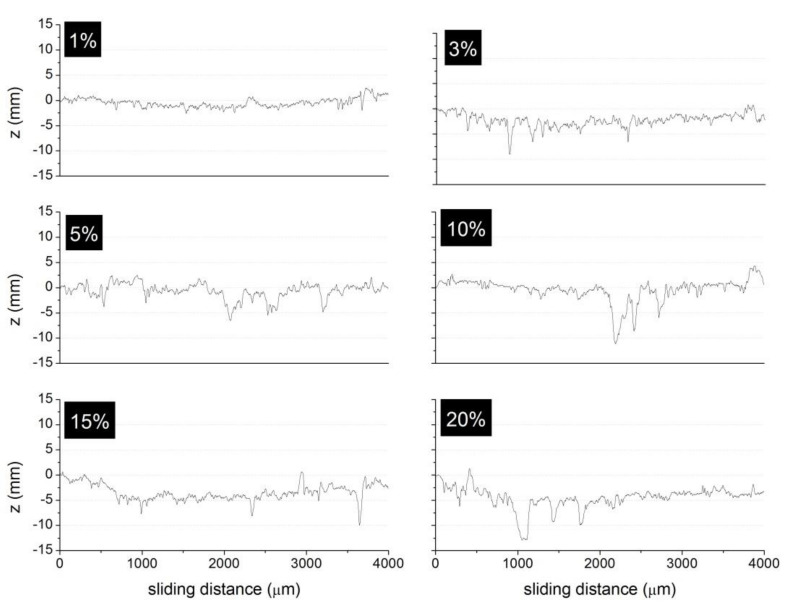
Surface profiles of the composite pad elements after the friction tests for 1%, 3%, 5%, 10%, 15%, and 20% wt. of A60 rubber particles.

**Figure 4 polymers-13-03371-f004:**
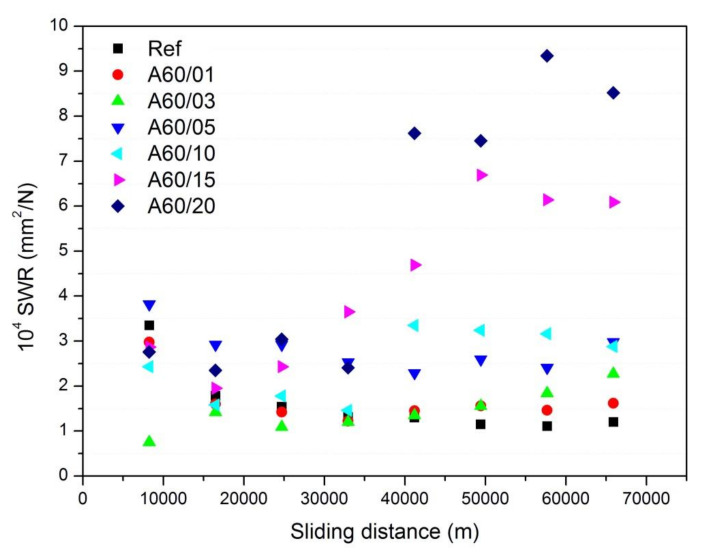
Evolution of SWR in the element pads containing different ELT rubber particles with the friction distance.

**Figure 5 polymers-13-03371-f005:**
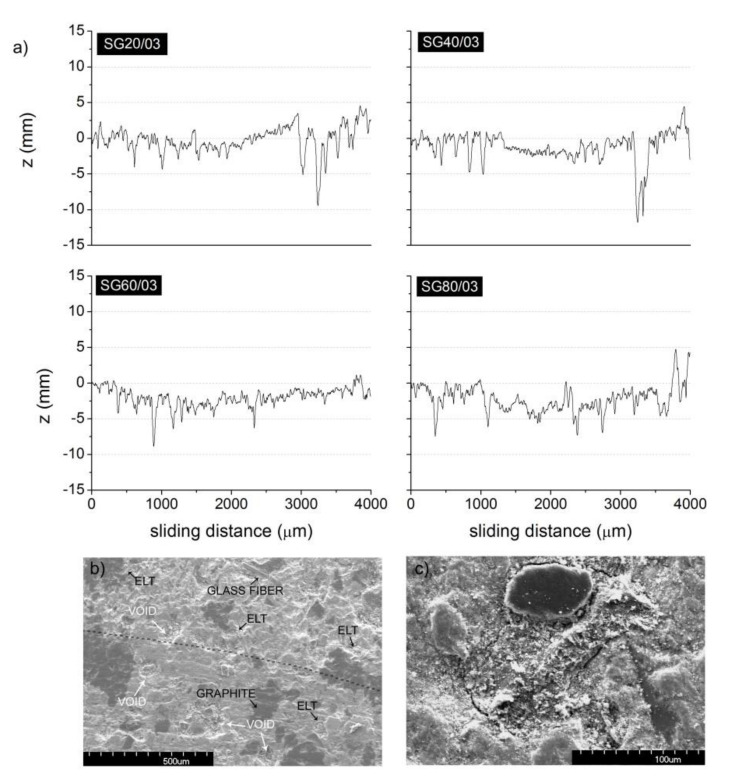
(**a**) Surface profiles after the friction test of the composite pad elements containing rubber particles C20, C40, A60, and A80. (**b**) SEM image of a tested brake pad C20/03 (dotted line indicates the limit of the pin-on-disk test) and (**c**) SEM image of a ELT rubber particle in C20/03 brake pad.

**Figure 6 polymers-13-03371-f006:**
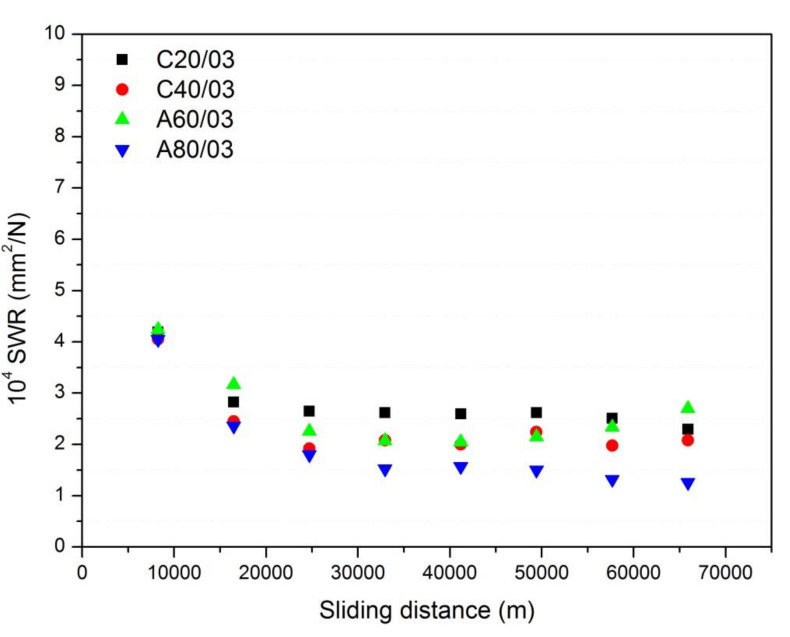
Evolution of *SWR* in the element pads containing ELT rubber particles of different sizes with the friction distance.

**Figure 7 polymers-13-03371-f007:**
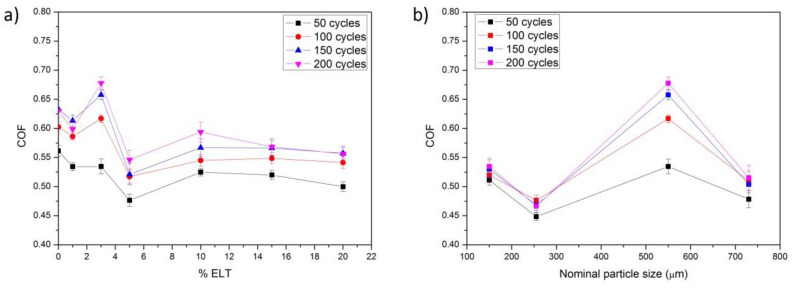
(**a**) COF of brake pads containing different concentrations of ELT rubber particles A60, and (**b**) COF of the brake pads containing 3% particles of different sizes.

**Figure 8 polymers-13-03371-f008:**
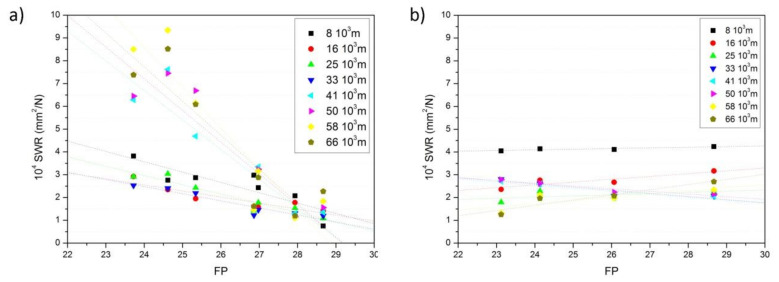
SWR and FP at the different braking distances (**a**) in the brake pads containing different concentrations of ELT and (**b**) in the brake pads containing ELT of different sizes.

**Table 1 polymers-13-03371-t001:** Composition of the reference pad elements (in the composite PAD elements, the ELT rubber particles are subsequently added in the specified proportions to this reference pad).

Material	Mean Particle Size (μm)	Density (g/cm^3^)	Concentration (% wt)
Phenolic resin	200	1.27	10
Graphite	1300	2.30	16
Glass Fiber	10	2.52	3
Steel particles	240	7.85	9
Barite	1.6	4.45	37
Alumina	900	3.85	25

**Table 2 polymers-13-03371-t002:** Characteristics of the ELT rubber samples used in this work.

Sample	Grinding Method	Comm. Name	Origin of ELT	Supplier	Mean Particle Size (μm)	Nominal Particle Size (μm)	Density (g/cm^3^)	Conc. in Pads(%wt)
A80	Ambient	Powder	50% truck tires/50% passenger car tires	Valoriza Medioambiente	450	650	1.16	3
A60					330	470	1.16	1, 3, 5, 10, 15, 20
C40	Cryogenic	GENAN 40 Mesh	100% truck tires	GENAN	170	320	1.16	3
C20		GENAN 80 Mesh			130	200	1.16	

**Table 3 polymers-13-03371-t003:** Theoretical and real densities (g/cm^3^) of the composite pad elements prepared with ELT rubber particles A60 (densities are given with ±0.05 error).

	Theoretical Density (g/cm^3^)	Real Density (g/cm^3^)	% Theoretical Density	Porosity (%)
Ref	3.13	2.90	92.73	7
A60/01	3.11	2.85	91.71	8
A60/03	3.07	2.78	90.60	9
A60/05	3.03	2.70	89.14	11
A60/10	2.93	2.50	85.31	15
A60/15	2.83	2.33	82.27	18
A60/20	2.73	2.20	80.47	20

**Table 4 polymers-13-03371-t004:** COF for brake pads prepared with ELT rubber particles A60 at different brake cycles and the corresponding wear rate values (W) at the end of the experiment.

	Number of Cycles				W (mm^3^)
	50	100	150	200	
Ref	0.561	0.602	0.633	0.630	0.115 +/− 0.082
A60/01	0.534	0.586	0.614	0.599	0.117 +/− 0.020
A60/03	0.538	0.616	0.658	0.678	0.141 +/− 0.111
A60/05	0.476	0.518	0.521	0.546	0.066 +/− 0.080
A60/10	0.524	0.659	0.567	0.593	0.092 +/− 0.066
A60/15	0.519	0.548	0.566	0.568	0.131 +/− 0.028
A60/20	0.484	0.541	0.557	0.556	0.164 +/− 0.080

**Table 5 polymers-13-03371-t005:** Weight loss (g) of brake pads with different rubber concentrations.

	Time Subjected to Braking Cycles (min)							
	60	120	180	240	300	360	420	480
Ref	0.016	0.017	0.022	0.025	0.031	0.033	0.037	0.046
A60/01	0.014	0.015	0.020	0.023	0.034	0.044	0.048	0.061
A60/03	0.019	0.029	0.031	0.038	0.047	0.059	0.075	0.099
A60/05	0.017	0.026	0.039	0.045	0.051	0.069	0.075	0.106
A60/10	0.010	0.013	0.022	0.024	0.069	0.080	0.091	0.095
A60/15	0.011	0.015	0.028	0.056	0.090	0.154	0.165	0.187
A60/20	0.010	0.017	0.033	0.035	0.138	0.162	0.237	0.247

**Table 6 polymers-13-03371-t006:** Theoretical and real densities (g/cm^3^) of the composite pad elements prepared with ELT rubber particles of different size (densities are given with ±0.05 error).

	Theoretical Density (g/cm^3^)	Real Density (g/cm^3^)	% Theoretical Density	Porosity (%)
C20/03	3.07	2.78	90.60	9
C40/03	3.07	2.79	90.93	9
A60/03	3.07	2.78	90.60	9
A80/03	3.07	2.75	89.62	10

**Table 7 polymers-13-03371-t007:** COF for brake pads prepared with the same concentration of ELT rubber particles and different particle size at different brake cycles and the corresponding wear rate value (W) at the end of the experiment.

	Number of Cycles				W (mm^3^)
	50	100	150	200	
C20/03	0.511	0.520	0.530	0.535	0.050 +/− 0.038
C40/03	0.448	0.477	0.468	0.467	0.113 +/− 0.079
A60/03	0.538	0.616	0.658	0.678	0.141 +/− 0.111
A80/03	0.478	0.509	0.504	0.517	0.075 +/− 0.080

**Table 8 polymers-13-03371-t008:** Weight loss (g) of brake pads with ELT rubber particles of different size.

	Time Subjected to Braking Cycles (min)							
	60	120	180	240	300	360	420	480
C20/03	0.020	0.027	0.038	0.05	0.062	0.075	0.084	0.088
C40/03	0.019	0.023	0.027	0.039	0.047	0.063	0.065	0.078
A60/03	0.019	0.029	0.031	0.038	0.047	0.059	0.075	0.099
A80/03	0.018	0.021	0.024	0.027	0.035	0.040	0.041	0.045

## Data Availability

Raw data available upon request to corresponding author.

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
