# Peer review of "Preparation and Properties of Sustainable Brake Pads with Recycled End-of-Life Tire Rubber Particles"

_polymers, 2021, doi:10.3390/polym13193371_

Round 1

Reviewer 1 Report

In the article "Preparation and properties of sustainable brake pads with recycled end-of-life tire rubber particles", the authors present a very interesting work in a clear and concise way. I strongly recommend the publication of this article in its present form.

Author Response

Dear Reviewer,

Many thanks for revising our manuscript. According to your comments, we have revised the text and some mistakes regarding to the English language were detected and corrected.

Thank you again for helping us to improve the quality of our manuscript

Sincerely Yours

A. Tamayo, on behalf of all co-authors

Reviewer 2 Report

The manuscript with title “Preparation and properties of sustainable brake pads with recy-cled end-of-life tire rubber particles” is interesting paper and it should be attractive to the readers of the Polymers journal.

I can recommend publication of the following manuscript with minor revision.

 The following are some comments:

Abstract

Authors should highlight the obtained results

Results and discussions

  1. Related to SEM analysis authors should provide more data. It is necessary to give images of investigated samples for which a particle size distribution is given. Also, the authors mentioned a specific surface and nowhere explained its influence on the desired properties of the material as well as the method of obtaining.
  2. Its remained unclear on the basis of which the authors chose sample A 60 to vary the % concentration of ELT rubber particles.
  3. The authors need to correct all technical errors, especially when it comes to the tittle of Figures and Tables (for example: rows:223, 239, 349—Table 33, figures 22?)
  4. The authors should provide a better

comparison of their results with the literature data. More precisely, the

authors did not clearly state in what way (percentage or measurement unit)

the results of their materials are more efficient and better in regard to

the parameters of other materials that have already been published.

Conclusions

Please, rewrite Conclusion. It is necessary to highlight main findings i.e. 

Author Response

Dear Reviewer,

We have received your comments and suggestions. Thank you for helping us to improve the quality of our manuscript. Please, find below a detailed response to your comments as well as the precise changes made on the manuscript to further clarify the questions.

Reviewer 2: Abstract. Authors should highlight the obtained results

Authors: The authors agree with the referee in the sense that the abstract should highlight the main results. Accordingly, the abstract already highlights the main results. We have included “It has been demonstrated” at the beginning of one of the sentences to emphasize that. The main findings are that the size of the particles are not the determinant factor on the friction wear mechanism and the adhesion of the particles enhances the long-life behavior of the brake pads, as already stated.

Reviewer 2: Related to SEM analysis authors should provide more data. It is necessary to give images of investigated samples for which a particle size distribution is given. Also, the authors mentioned a specific surface and nowhere explained its influence on the desired properties of the material as well as the method of obtaining.

Authors: We agree with the reviewer that in the manuscript it was not specified at which sample correspond each image. In the revised version, we have clearly specified the samples selected for the images (C20 and A80). In addition, as the reviewer suggested, we have included in each case a low magnification image where the different particle sizes can be appreciated and afterwards confirmed in the particle size distributions, as determined by Dynamic Light Scattering. Moreover, in the introduction section, the authors provided more details about the two grinding mechanism which have been applied to obtain the studied ELT rubber particles, which justify the observation of the different surface properties.

Reviewer 2: Its remained unclear on the basis of which the authors chose sample A 60 to vary the % concentration of ELT rubber particles.

Authors: The selection of the A60 particles was primarily based on two reason: First of all, the size of the particles was in the middle of all the studied particles. However, the main reason for selecting these particles is because it is known that the cohesion forces between particles depend not only on their concentration but on their polydispersity. In  highly polydispersed systems, the small particles fills the voids of the large particles, however, the best contact between particles is obtained when using particles of uniform radius (i.e. low polydispersed).

In the manuscript, we have including the following paragraph where it is justified the selection of the A60 particles in the concentration study of the frictional properties of the brake pads:

It is known that cryogenic grinding is more effective in producing fine particles than ambient grinding [32], as it can be deduced from the particle size distributions shown in Figure 1. In these particle size distributions, it has been calculated the size span which is defined as (d90 – d10)/(d90 + d10), being  d10 and d90 the size of the particles those dimensions where the 10% wt. and 90% wt. of the PSD (notice that d90 corresponds to the nominal value). The encountered size span values are 0.57, 0.60, 0.50 and 0.42 for C20, C40, A60 and A80, respectively, indicating that these particles obtained through ambient grinding are less polydisperse than the ones obtained by the ambient grinding procedure. According to Voivret et al [33], polydispersity of a granular media do not affect the shear strength but it dominates the adhesion forces between particles. Additionally, the concentration of ELT rubber particles as well as their mean size play have been demonstrated to play a key role on the tribological properties of the composite brake pads. Chang et al. [19] reported a decrease of friction level with the particle size in brake pads prepared with 10% rubber particles, whereas Liu et al. [20]  stated that by employing up to 5% of rubber nanoparticles, an improvement of the friction properties with the size of the particles was obtained. To study the effect of the concentration in the brake pads, it has been selected the particles A60, possessing the medium size span (s = 0.50) to minimize the effect of the different packing density of the ELT rubber particles in the brake pads attributed to the different particle polydispersity. With these considerations, and based on the data presented in Figure 2 b and collected in Table 4 and Table 7, we can extract that although the normal behavior is an increase of the COF with the number of braking cycles, this increase in the COF is most noticed in the brake pads containing 3% ELT rubber particles and when using the particles with the mentioned size (A60), as shown in Figure 7.

Reviewer 2: The authors need to correct all technical errors, especially when it comes to the tittle of Figures and Tables (for example: rows:223, 239, 349—Table 33, figures 22?)

Authors: The technical errors were also detected in the version of the manuscript which was available for download, but not in the Author´s version nor in the revised version prior to confirm submission. We are sorry for that. We have removed all the cross-references in an attempt to avoid the re-occurrence of this issue.

Reviewer 2: The authors should provide a better comparison of their results with the literature data. More precisely, the authors did not clearly state in what way (percentage or measurement unit) the results of their materials are more efficient and better in regard to the parameters of other materials that have already been published.

               Authors: According to the reviewer´s suggestion, we have included some new discussions about the values obtained and compared with those found by some other authors. The reviewer will be able to find in the text the precise changes made (highlighted in the text). Addittionally, in this response letter, we have extracted some of the new sentences added:

P18: Liew et al. [34] prepared new brake pad material where the harmful asbestos component was replaced and compared the COF of the material with a commercial brake pad. In all the cases, the COF barely reached 0.5, which is almost the minimum value of the brake pads containing the A60 particle. Contrary to that Liew et al. found, the COF increase with the sliding distance because of the different abrasion mechanism (plastic deformation attributed to the presence of the rubber particles).

P19: The obtained SWR values are between 10-4 mm3/N for the pads containing the A80 particles and at high sliding distances to about 9 10-4 mm3/N for the brake pads contining the A60 particles at the maximum concentration. These vaues fall in the range of some brake pads containing natural recycled particles [39] where it has been reported a SWR value of 7.22 10-4 mm2/N under a load application of 7.5 N, but they are higher than the composite pads filled with abrasive ceramics possessing COF of about 0.7 and SWR values below 4 10-4 mm2/N at 20 N load [40].   

P20: Since the same concentration of particles are used, the increase of SWR with the FP should be attributed to different adhesion forces because of the different grinding procedures followed for obtaining the particles and therefore to a different particle-matrix interaction. This result is similar to the studies carried out by  Zhang et al. [41] who demonstrated that the reaction between the different components of the brake pads because of the temperature increase at high friction cycles might contribute to the occurrence of the different abrasion mechanism found as the sliding distance increases.

Reviewer 2: Conclusions Please, rewrite Conclusion. It is necessary to highlight main findings

Author: Accordingly to the reviewer suggestion, we have reorganized the conclusion section and some new sentences have been added to highlight the main findings of the manuscript

Sincerely yours

A. Tamayo, on behalf of all the co-authors

Reviewer 3 Report

Please use the journal template for the manuscript.

"Table 2 Composition of the composite pad elements (ELT rubber particles are subsequently added in the specified proportions to this composition). " - If ELT particles are added to this composition, then concentrations of ELT particles are not in wt%, so correct it.

Is actually mean particle size of glass figer 10 micrometers? Seems unusual for glass fibers. Also, how it is fiber then?

Were ELT particles obtained from passenger or truck tires? Also were both producers using similar compositions?

"In the formulation of the actual composite brake elements, the authors also emphasize, that commercially available brake pads contain more than 30 constituents. Nevertheless, only seven ingredients have been employed here in order to clearly identify the effect of the ELT rubber particle interactions with the remainder elements. In the following sections, the influence of concentration and particle size of ELT rubber in the friction and wear properties of high COF brake pad composite elements are described." - if theere are over 30 constituents commercially then how Authors know that ELT particles will only interact with applied 7 ones not the others?

Please present the images of prepared materials. Porosity values indicate that Authors obtained rather foams than solid materials.

Please present the formula for calculating theoretical density of composite pads. Quick calculations show that theoretical density of Ref sample should equal around 3.13 g/cc.

" As indicated in the experimental section, during the whole 20 seconds of the test experiment, the rotation speed is raised from 0 to 2000 within the first 2 seconds and then it is maintained for other 18 seconds until stop. During the first 10 seconds of the experiment, the friction force increases from 0 to 2 N and afterwards the rotation speed is maintained but the friction force is increased to 7 N to simulate braking during other 10 seconds. Finally, in the last 3 s, the applied force decreases to 2 N and the speed is back to 0 rpm. " - such information should be presented in the experimental section, no need for repetition and artificial elongation of the discussion.

Please enhance Figure 2, it is hardly visible.

". It is nevertheless clear that the material removal increases with the rubber concentration in the pad element, as a result of the low wear resistance of such particles." - please show the surface images, the best with EDX analysis showing that actually ELT particles were removed not the other ones. 

"It is observed that the wear volumes increase with the rubber concentration in the pad element, as corresponds to a low interfacial bonding between the rubber particles and the binding resin" - as far as I am concerned 0.066 and 0.092 is less than 0.115, the this statement is not true. Also 0.131 is less than 0.141. 

"This phenolic resin used as binder presents a better surface interaction with the inorganic and metal components forming part of the composition of the brake pad element than with the rubber particles." - please provide any proof for such statement.

Which one is true:

  • "In the prepared pad elements, it is observed a decrease of the W values for rubber concentrations between 5 and 10%"

or

  • "It is observed that the wear volumes increase with the rubber concentration in the pad element"

??

Only couple lines between these two contradictory statements.

If the best results were noted for ELT particles content of 5 or 10 (not %wt), then why 3 was selected for comparison of different types?

How comes that the W is higher for C40 than C20 and SWR is lower?

Author Response

Reviewer: Please use the journal template for the manuscript.

Authors: We have used the journal template for the mansucript

Reviewer:  "Table 2 Composition of the composite pad elements (ELT rubber particles are subsequently added in the specified proportions to this composition). " - If ELT particles are added to this composition, then concentrations of ELT particles are not in wt%, so correct it.

Authors: To avoid confusion, we have modified the table caption and we have specified that this composition corresponds to the reference pad element and the ELT rubber particles were subsequently added to this reference composition in the specified wt%

 Reviewer:  Is actually mean particle size of glass figer 10 micrometers? Seems unusual for glass fibers. Also, how it is fiber then?

Authors: In the revised version of the manuscript, we have specified that we have used chopped glass fibers with a men size of 10 micrometers

 Reviewer: Were ELT particles obtained from passenger or truck tires? Also were both producers using similar compositions?

Authors: The Reviewer has addressed an excellent point which deserves to be considered in the investigations. It is true that passenger car tires and truck tires possess different amounts of natural and synthetic rubbers. The most common natural rubber employed for tire fabrication consists on a polymer of isoprene (2-methyl-1,3-butadiene) whereas in the case of synthetic rubbers, styrene-butadiene rubbers are the most widely used. In general, passenger tires are fabricated with about 27% synthetic rubber and 14% natural rubber whereas truck tires are fabricated with about 14% synthetic rubber and 27% natural rubber. This of course could affect the adherence of the rubber particles to the matrix into the composite pad element. In fact, this circumstance has been addressed in the introductory part of our manuscript but not studied in deep in this work. We consider of high importance that point and it will be the focus of future research.

However, apart from the origin of the tire (passenger or truck), the distribution of materials is different in each part of the tire. A typical tread cap consists of natural rubber, synthetic rubber and butadiene rubber, the inner liner is mainly composed of butyl rubber, whereas a typical sidewall formulation contains a blend of Natural Rubber, Butadiene Rubber and Reclaim Rubber (rubber recovered from vulcanized scrap rubber). Indeed, in our manuscript, we used a complex mixture obtained from all the different tire parts (inner liner, tread, sidewalls), as specified in the experimental section. It was not considered the percentage amount of each part in the final mixture but, as we recognized above, this effect could be also the subject of future research. To resume, we did not consider the origin of the tires since, in our opinion, there are several parts, apart from the origin of the tire that could affect the chemical composition of the ELT rubber particles. In our work, we just focused our attention on the different surface properties obtained by the different grinding methods. However, we recognize the other points worth to be studied.

Reviewer: "In the formulation of the actual composite brake elements, the authors also emphasize, that commercially available brake pads contain more than 30 constituents. Nevertheless, only seven ingredients have been employed here in order to clearly identify the effect of the ELT rubber particle interactions with the remainder elements. In the following sections, the influence of concentration and particle size of ELT rubber in the friction and wear properties of high COF brake pad composite elements are described." - if theere are over 30 constituents commercially then how Authors know that ELT particles will only interact with applied 7 ones not the others?

Authors: The reviewer is right that the ELT particles could interact with any of the other ingredients in the formulation of the brake pads. Each trade name uses its own composition and it would be technologically inviable to test each single composition in the market. We have selected the 7 components that are common in the commercial brake pads and in an average percentage amount.

Reviewer: Please present the images of prepared materials. Porosity values indicate that Authors obtained rather foams than solid materials. Please present the formula for calculating theoretical density of composite pads. Quick calculations show that theoretical density of Ref sample should equal around 3.13 g/cc.

Authors: The two questions have been answered together. Thanks to the reviewer, we have noticed an error in the calculations of the theoretical densities. Right values are now added in the revised version of the manuscript. Accordingly, the porosity values are rather lower than the ones reported before. The authors really apologize for not noticing the mistake before and are really grateful to the reviewer for pointing out that. Now, the porosity values are below 20% in all the cases and this porosity might be due to the shrinkage of the ELT rubber particles occurring during the processing. The highest is the amount of ELT particles, the highest is the porosity amount. Similarly, we have corrected the density values of Table 6. Despite this error, the actual conclusions are the same (no influence of the particle size on the density but of its concentration instead).

Reviewer: " As indicated in the experimental section, during the whole 20 seconds of the test experiment, the rotation speed is raised from 0 to 2000 within the first 2 seconds and then it is maintained for other 18 seconds until stop. During the first 10 seconds of the experiment, the friction force increases from 0 to 2 N and afterwards the rotation speed is maintained but the friction force is increased to 7 N to simulate braking during other 10 seconds. Finally, in the last 3 s, the applied force decreases to 2 N and the speed is back to 0 rpm. " - such information should be presented in the experimental section, no need for repetition and artificial elongation of the discussion.

Author: According to the reviewer suggestion, this information has been removed from the result section.

Reviewer: Please enhance Figure 2, it is hardly visible.

               Author: We have enhanced Figure 2 to make it clearer

Reviewer: ". It is nevertheless clear that the material removal increases with the rubber concentration in the pad element, as a result of the low wear resistance of such particles." - please show the surface images, the best with EDX analysis showing that actually ELT particles were removed not the other ones.

Author: The authors acknowledge the reviewer this valuable comment. In the revised version of the manuscript, we have included the SEM images of the eroded area as Figure 5b. In addition, Figure 5c presents the poor interface between the ELT rubber particles and the brake pad matrix. The following paragraph has been included in the revised version of the manuscript addressing the referred issue:

“In Figure 5 b it is shown the SEM image of the surface of the brake pad labeled C20/03 after being subjected to the pin-on-disk test. For clarification, we have highlighted with a dotted line the limit of the test (the bottom part of the image corresponds to the eroded area). There, it is observed some graphite particles, as planar, dark and big particles, glass fibers and ELT rubber particles present in both parts of the image. In the eroded area, it is possible to observe some voids appearing because of the elimination of particles with the consequent debris formation as well as some remainders of the tribolayer formed during the test. The size of the voids is about 100-200 m which is approximately the size of the ELT rubber particles. In the images, the identification of the ELT rubber particles in the brake pad elements has been realized by recognizing these particles with a weak or even hollow interface, which is the origin of the porosity in the pad elements. In Figure 5 c it is presented the high resolution SEM image of a ELT rubber particle partially covered with a planar graphite particle where it can be distinguished the poor interface between the matrix and the ELT rubber particle.”

Additionally, according to the reviewer suggestion, we have carried out the EDX analysis of the abraded area and the brake pad elements before being tested. Unfortunately, no significant differences could be observed. Taking into account that the main components of ELT are C and S, and both elements are already present on the composition of the different elements of the brake pads, the encountered differences fall within the error of the instrument so no valid conclusions could be extracted out of the results.

 Reviewer: "It is observed that the wear volumes increase with the rubber concentration in the pad element, as corresponds to a low interfacial bonding between the rubber particles and the binding resin" - as far as I am concerned 0.066 and 0.092 is less than 0.115, the this statement is not true. Also 0.131 is less than 0.141.

Authors: We have clarified that the general trend is an increase of the wear rate with the ELT concentration but, as Mutlu et al. reported, there might occur a beneficial effect at particle concentrations of about 5-10% which exerts a punctual decrease in the wear rate.

 Reviewer: "This phenolic resin used as binder presents a better surface interaction with the inorganic and metal components forming part of the composition of the brake pad element than with the rubber particles." - please provide any proof for such statement.

Author: A new reference has been added in the text regarding the different interactions of the phenolic resin with the components of the braking pads: Zhao, Xiaoguang; Ouyang, Jing; Tan, Qi; Tan, Xiumin; Yang, Huaming. Interfacial characteristics between mineral fillers and phenolic resin in friction materials. Materials Express 10(1) (2020) 70-80

Reviewer: Which one is true: "In the prepared pad elements, it is observed a decrease of the W values for rubber concentrations between 5 and 10%" or "It is observed that the wear volumes increase with the rubber concentration in the pad element"?? Only couple lines between these two contradictory statements.

Author: For this question, the Authors refers to their previous response: We have clarified that the general trend is an increase of the wear volume with the rubber concentration, but at 5 and 10% rubber particles, similarly to that Mutlu found, a decrease in the wear rate was detected, which in turn was dependent on the porosity and density of the final materials.

Reviewer: If the best results were noted for ELT particles content of 5 or 10 (not %wt), then why 3 was selected for comparison of different types?

Author: The authors acknowledge the reviewer for pointing out this question. In the revised version of the manuscript, there were referenced the works of Chang et al. and Liu et al. which reported decreased friction levels with the particle size at particle concentrations below 10 and 5%, respectively. We used 3% ELT rubber particles to evaluate the effect of the particle size to get rid of any anomalies caused by inhomogeneous distributions of the particles in the brake pad matrix.

Reviewer: How comes that the W is higher for C40 than C20 and SWR is lower?

Author: SWR and W are calculated taking into account different parameters. Whereas W is calculated from the worn volume, the SWR is calculated from the remove mass. The results obtained in our work are in accordance with the work of Chang et al who found a decrease in the SWR values as the particle size increase, an attributed this behavior to the different formation of wear debris. This is already addressed in the manuscript.

Reviewer 4 Report

Dear Authors,

This manuscript is quite interesting. It also covers the topic of tire recycling, which is good for the environment. There are, however, some shortcomings in the manuscript that need to be corrected.

Detailed comments below:

I must admit that instead of the chapter name "Experimental" I prefer "Materials and methods"

The methodology lacks information on the microscopic analysis of SEM. All experiments and scientific equipment must be included and described in the methodology.

The methodology lacks information on the number of repetitions of the experimental results.

Table 1. All tables should be prepared in accordance with the journal's guidelines.

The porosity tests are not described in the test methodology. This has to be added to the methodology. In addition, porosity is typically the number of pores per unit area / volume. In table 3, the porosity is in%. You have to explain it at work. If this is a mistake, correct it.

Figure 5. In the description of the figure, add what "Z" means on the y axis.

Figure 4, 6, 7, 8. If possible, error bars should be used against the data points.

In addition, the research methodology should include information about the basic statistical analysis that was used to analyze the research results. Unfortunately, this is missing from this manual. The statistical analysis is: Anova, error bars, SD. In the case of your selected results, such a simple analysis may be more than enough.

"Discussion": This subsection is a further part of the research results. I think it is better to apply one chapter "Results and Discussion".

Conclusion: One more general conclusion should be added. Looking more ahead.

Author Response

Reviewer: I must admit that instead of the chapter name "Experimental" I prefer "Materials and methods"

Authors: According to the reviewer suggestion, we have change the name of the section to “Materials and Methods”

Reviewer: The methodology lacks information on the microscopic analysis of SEM. All experiments and scientific equipment must be included and described in the methodology.

Authors: The information about the instrument and methodology for observation (gold sputtering) has been included in the revised version of the manuscript.

Reviewer: The methodology lacks information on the number of repetitions of the experimental results.

Authors: In the experimental section, it indicated that all specimens were prepared in duplicate.

Reviewer: Table 1. All tables should be prepared in accordance with the journal's guidelines.

Authors: We have employed the Journal Template to prepare our manuscript. We have revised the template and correct the formatting errors that were detected.

Reviewer: The porosity tests are not described in the test methodology. This has to be added to the methodology. In addition, porosity is typically the number of pores per unit area / volume. In table 3, the porosity is in%. You have to explain it at work. If this is a mistake, correct it.

Author: We have specified that the porosity (in %) was calculated from the diference between the theoretical and real density, as determined by the immersion method.

Reviewer: Figure 5. In the description of the figure, add what "Z" means on the y axis.

Author: In the revised version of the manuscript we have specified that the z value corresponds to the surface profile, in mm

Reviewer: Figure 4, 6, 7, 8. If possible, error bars should be used against the data points. In addition, the research methodology should include information about the basic statistical analysis that was used to analyze the research results. Unfortunately, this is missing from this manual. The statistical analysis is: Anova, error bars, SD. In the case of your selected results, such a simple analysis may be more than enough.

Author: Unfortunately, the samples were made in duplicate, as mentioned in the experimental section, thus error bars cannot be calculated. In Figure 7, however, we have specified that the error bars are calculated from the SD values of all the braking cycles in steps of 50 cycles.

Reviewer: "Discussion": This subsection is a further part of the research results. I think it is better to apply one chapter "Results and Discussion".

Author: We are sorry of being in slight discrepancy with the reviewer. We have differentiated the data obtained from the test as just “Results” and the processing of the data to extract the conclusions takes part of the discussion section. We recognize that the structure followed to present our results might not like all the readers, however, from our point of view, it was the one which present our research in the more comprehensive manner.

Reviewer: Conclusion: One more general conclusion should be added. Looking more ahead.

Author: At the end of the conclusion section, we have added a couple of sentences highlighting the general conclusion.

Round 2

Reviewer 3 Report

"

Reviewer:  "Table 2 Composition of the composite pad elements (ELT rubber particles are subsequently added in the specified proportions to this composition). " - If ELT particles are added to this composition, then concentrations of ELT particles are not in wt%, so correct it.

Authors: To avoid confusion, we have modified the table caption and we have specified that this composition corresponds to the reference pad element and the ELT rubber particles were subsequently added to this reference composition in the specified wt%"

There is still a mistake. Table 2 reports the composition of reference sample  and it adds up to 100 wt%. Then, Authors are claiming to add 3 or more wt% of rubber. But if so, then total amount of components increase and for example loading of glass fiber is not 3 wt% anymore but 3/120 = 2.5 wt% for A60 content of 20 wt%. Therefore, I do not think the rubber was added in wt%. Such a basic mistakes should be clarified definitely.

" Reviewer: Were ELT particles obtained from passenger or truck tires? Also were both producers using similar compositions?

Authors: The Reviewer has addressed an excellent point which deserves to be considered in the investigations. It is true that passenger car tires and truck tires possess different amounts of natural and synthetic rubbers. The most common natural rubber employed for tire fabrication consists on a polymer of isoprene (2-methyl-1,3-butadiene) whereas in the case of synthetic rubbers, styrene-butadiene rubbers are the most widely used. In general, passenger tires are fabricated with about 27% synthetic rubber and 14% natural rubber whereas truck tires are fabricated with about 14% synthetic rubber and 27% natural rubber. This of course could affect the adherence of the rubber particles to the matrix into the composite pad element. In fact, this circumstance has been addressed in the introductory part of our manuscript but not studied in deep in this work. We consider of high importance that point and it will be the focus of future research.

However, apart from the origin of the tire (passenger or truck), the distribution of materials is different in each part of the tire. A typical tread cap consists of natural rubber, synthetic rubber and butadiene rubber, the inner liner is mainly composed of butyl rubber, whereas a typical sidewall formulation contains a blend of Natural Rubber, Butadiene Rubber and Reclaim Rubber (rubber recovered from vulcanized scrap rubber). Indeed, in our manuscript, we used a complex mixture obtained from all the different tire parts (inner liner, tread, sidewalls), as specified in the experimental section. It was not considered the percentage amount of each part in the final mixture but, as we recognized above, this effect could be also the subject of future research. To resume, we did not consider the origin of the tires since, in our opinion, there are several parts, apart from the origin of the tire that could affect the chemical composition of the ELT rubber particles. In our work, we just focused our attention on the different surface properties obtained by the different grinding methods. However, we recognize the other points worth to be studied."

The issue of ELT origin is a simple question and has to be resolved in order to go forward with the paper, because Authors have to present what type of rubber particles they used.

Author Response

Reviewer 3 Round 1:  "Table 2 Composition of the composite pad elements (ELT rubber particles are subsequently added in the specified proportions to this composition). " - If ELT particles are added to this composition, then concentrations of ELT particles are not in wt%, so correct it.

Reviewer 3 Round 2:  There is still a mistake. Table 2 reports the composition of reference sample  and it adds up to 100 wt%. Then, Authors are claiming to add 3 or more wt% of rubber. But if so, then total amount of components increase and for example loading of glass fiber is not 3 wt% anymore but 3/120 = 2.5 wt% for A60 content of 20 wt%. Therefore, I do not think the rubber was added in wt%. Such a basic mistakes should be clarified definitely.

Authors: The reviewer is absolutely right in affirming that varying the amount of rubber will inherently modify the relative amount of the other elements. We have modified the experimental section to clarify this issue. Now, in the modified version of the manuscript, it has been specified that the components included in Table 1 (notice that we have also change the position of the two tables, Table 1 and Table 2) forms part of the reference material (0 % ELT). To process the composite materials, the ELT was incorporated to the mixture in the concentrations specified in Table 2. Accordingly, in Table 2, it is also specified that the referred concentration is the wt % in the pad.

With these modifications, the author´s expectation is that the Materials and Methods section is clearer and more reproducible than in the previous version of the manuscript.

Reviewer 3 Round 1:  Were ELT particles obtained from passenger or truck tires? Also were both producers using similar compositions?

Reviewer 3 Round 2:  The issue of ELT origin is a simple question and has to be resolved in order to go forward with the paper, because Authors have to present what type of rubber particles they used.

Authors: According to the reviewer suggestion, the origin of the ELT particles was now specified in Table 2

Reviewer 4 Report

The authors complied with most of the comments. I accept the explanations and accept the corrections made.
Regards

Author Response

The authors acknowledges the reviewer his/her effort to improve the quality of our manuscript